# ProAKAP4 Semen Concentrations as a Valuable Marker Protein of Post-Thawed Semen Quality and Bull Fertility: A Retrospective Study

**DOI:** 10.3390/vetsci9050224

**Published:** 2022-05-06

**Authors:** Marta Dordas-Perpinyà, Nicolas Sergeant, Isabelle Ruelle, Jean-François Bruyas, Frédéric Charreaux, Sandrine Michaud, Sara Carracedo, Jaime Catalán, Jordi Miró, Maryse Delehedde, Lamia Briand-Amirat

**Affiliations:** 1Oniris, Nantes-Atlantic College of Veterinary Medicine, 44300 Nantes, France; martadordas@gmail.com (M.D.-P.); isa.ruelle@hotmail.fr (I.R.); jean-francois.bruyas@oniris-nantes.fr (J.-F.B.); sandrine.michaud@oniris-nantes.fr (S.M.); 2Equine Reproduction Service, Department of Animal Medicine and Surgery, Faculty of Veterinary Sciences, Autonomous University of Barcelona, 08193 Cerdanyola del Vallès, Spain; dr.jcatalan@gmail.com (J.C.); jordi.miro@uab.cat (J.M.); 3U1172 LilNCog—Lille Neuroscience & Cognition, CHU Lille, Inserm, University Lille, 59000 Lille, France; nsergeant@4biodx.com; 4SPQI S.A.S, 82 rue Jeanne d’Arc, 59000 Lille, France; carracedovicentesara@gmail.com (S.C.); mdelehedde@4biodx.com (M.D.); 5EVOLUTION XY, 35538 Noyal sur Vilaine, France; frederic.charreaux@evolution-xy.fr

**Keywords:** proAKAP4, AKAP4, cryopreservation, motility, fertility, bull

## Abstract

Functional sperm quality markers to predict bull fertility have been actively investigated. Among them, proAKAP4, which is the precursor of AKAP4, the main structural protein in the fibrous sheath of spermatozoa; appears to be promising, especially since spermatozoa lacking AKAP4 expression were shown to be immotile, abnormal, and infertile. In this study, the objective was to evaluate proAKAP4 concentration values with the classic sperm motility descriptors and fertility outcomes (NRR at 90 days) in post-thawed conditions of 10 bulls’ semen. ProAKAP4 expression was confirmed by Western blotting and proAKAP4 concentrations were determined by ELISA. Variations in proAKAP4 concentrations were observed independently of the motility sperm descriptors measured using computer-assisted semen analysis (CASA). A ProAKAP4 concentration of 38.67 ± 8.55 ng/10 million spermatozoa was obtained as a statistical mean of all samples. Threshold values of proAKAP4 were then determined between 19.96 to 96.95 ng/10 million spermatozoa. ProAKAP4 concentrations were positively correlated with progressive motility and the linearity coefficient. The sperm showing the lowest progressive motility were the samples exhibiting proAKAP4 concentrations below 20 ng/10 million spermatozoa. Furthermore, proAKAP4 concentrations were significantly higher in bulls with a higher NRR in the field. Our results demonstrate a correlation between the semen concentration of proAKAP4 and NRR-90d (*p* = 0.05) in post-thawed bull semen, highlighting the potential of proAKAP4 as a predictive marker of bull fertility.

## 1. Introduction

Genetic improvement in dairy farming has long been motivated by increasing milk production and improving milk fat and protein levels, at the expense of zootechnical and reproductive performance. However, the latter is necessary for profitable dairy farming. Reproductive failure consequences include increased calving intervals and decreased milk production, both leading to a loss of productivity and significantly increased economic costs for breeders [1]. Despite the male reproductive ability to fertilize having a significant economic impact on the dairy industry, the relationship between production and male fertility is not well understood [2,3]. The success of artificial insemination is affected by multiple variables, including the age of the bull, sperm quality of the frozen-thawed semen [4,5,6,7], the breeder’s technical skills, and the genetics of the herd, but also the quality criteria or standards of the semen proposed by the sire’s center [8].

Bull fertility is a complex multifactorial phenomenon and accurate predictions require biological and statistical information [9]. Bull semen, which can allow genetic improvement, must be as fertile as possible to improve breeding rates [10]. Cattle fertility has a low heritability and the selection of bulls by their progeny’s fertility is difficult [3,11,12]. For sperm production, bulls are selected first on the basis of their genetic ability and fertility, and then the quality of the semen is tested for each pack of produced straws; sperm must be of good quality to maintain its fertility [13]. Field fertility and sperm quality parameters have been often associated [7]. However, even if classical measures of semen quality like sperm concentration, motility, and viability are used to characterize ejaculate, the studies on their association with pregnancy rates are inconsistent [11,14]. New technologies, like metabolomics, proteomics, and genomics studies are conducted and motivated by the research of new male fertility biomarkers [7].

Classically, motility descriptors have been used to estimate sperm quality and fertility. Sperm motility, essential for the spermatozoa to reach the site of fertilization, is considered an essential sperm quality parameter of semen fertility. Motility, provided by the sperm tail, is under the control of a complex molecular and structural machinery, among which A-kinase anchor protein 4 (AKAP4) was shown to play an important role [15,16,17].

Structurally, AKAP4 protein is the most abundant constitutive protein of the sperm fibrous sheath being part of the principal piece of the flagellum in all mammals including bulls [18,19]. This scaffold protein is instrumental in regulating sperm flagellum motion and an AKAP4 deletion results in a defective fibrous sheath formation and flagellar dysplasia, resulting in infertility due to the loss of motility and sperm abnormalities [18,20,21]. Furthermore, AKAP4 has been described as a marker of mature sperm [22]. Interestingly, a strong negative correlation between the proAKAP4 concentrations and the percentage of sperm tail abnormalities has been recently described in bull semen [23].

Structurally, AKAP4 is synthesized as a precursor protein named proAKAP4 and the release of a sequence peptide called prodomain provides the mature AKAP4 [17]. Both AKAP4 and proAKAP4 are found in mature sperm [16,24,25] and proAKAP4 has been considered an objective sperm quality and fertility biomarker [16,26]. In humans, AKAP4 is clearly downregulated in infertile patients with asthenozoospermia [27,28]. Evidence of the role of AKAP4 in bull fertility has increased over the years [16,29,30,31,32]. AKAP4 transcripts were identified among the top five validated transcripts, and significantly increased in high compared to low quality bull semen [30]. By testis transcriptome profiling, the AKAP4 gene was found to be drastically downregulated in sterile hybrid male cattle yaks due to spermatogenic arrest [31]. The proAKAP4 concentrations were recently described to be correlated with the total and progressive motility of spermatozoa in bulls [23].

In the field, ejaculate quality is often crudely and subjectively assessed using a microscope or objectively analyzed using expensive computer-assisted semen analysis (CASA) equipment. Using the Bull 4MID^®^ Kit (4BioDx, Lille, France) for ProAKAP4 analysis may instead offer a less expensive and a more accurate and objective analysis of semen quality.

The aim of this work was then to evaluate the relationships between the sperm proAKAP4 levels measured by ELISA (as an in vitro functional assays), the conventional semen analysis, and fertility indicators (non-return rate at 90 days) in post-thawed conditions using commercially available straws.

## 2. Materials and Methods

### 2.1. Sperm Descriptors Analysis in Post-Thawed Straws

Semen samples were collected from 10 Prim’Holstein bulls aged between 17 months and 7 years old; each bull was collected 4 times on different days, and one straw of each collection was taken (*n* = 40). Straws were obtained from a semen production center (Evolution XY, Blain, France). Semen samples were prepared using the Optidyl medium (IMV Technologies, L’Aigle, France) to reach a final concentration of 80 million spermatozoa/mL (M spz/mL), packaged and frozen in 0.25 mL straws (IMV Technologies, L’Aigle, France). The straws were taken in liquid nitrogen storage tanks from the bovine reproduction center Evolution XY (Blain, France). Each straw was thawed by immersion in a water bath at 37 °C for 50 s. The sperm was kept in an incubator at 37 °C and protected from light. Thawed semen was quarter-fold diluted in Easy Buffer B medium. Sperm parameters (concentration, motility) were assessed using Leja cells (IMV Technologies, L’Aigle, France) in a computer-assisted semen analysis (CASA) (IVOS version 12.0, Hamilton Thorne Research, Beverly, MA, USA). Obtained motility descriptors are total motility (TM), progressive motility (PR), curvilinear velocity (VCL), straight-line velocity (VSL), average path velocity (VAP), linearity (LIN), straightness (STR), amplitude of lateral head displacement (ALH), and beat-cross frequency (BCF). The CASA setup data were number of sequences: 30, number of fields per sequence and second: 60, percentage of progressive motility: % of spermatozoa with VAP > 30 µm/s and STR > 80%; minimum total motility threshold: VAP > 20 µm/s; cell size detector: 3 pixels. The analysis was repeated three times in ten different fields. The same measures were made 10 min later, keeping the samples in the water bath at 37 °C.

### 2.2. Semen Sample Preparation

Aliquots of 0.2 mL of post-thawed semen were frozen at −20 °C, sealed with paraffin, and shipped overnight in dry ice to INSERM UMRS-1172 research unit in Lille (France) for the proAKAP4 analyses and Western blot. Before separating spermatozoa from the cryopreservative, samples were thawed by immersion in a water bath at 37 °C for one minute.

### 2.3. Separation of Spermatozoa from the Cryopreservative

For the separation of spermatozoa and cryopreservative, a volume of 50 µL of semen was pipetted in a 1.5 mL Eppendorf tube and added to 150 µL of PBS and centrifuged at 2000 rpm at 20 °C for 10 min. The supernatant over the spermatozoa pellet was recovered with a 200 µL pipette. The supernatant was either added to one volume of PBS with 2% SDS and sonicated on the ice at 22 kHz, 15 Watt for 30 s, or with 100 µL of the 4MID^®^ Kit Bull Lysis Buffer (4BioDx, Lille, France). The spermatozoa pellet was added to 250 µL of PBS with 2% SDS and further sonicated on the ice at 22 kHz and 15 Watt for 30 s. The spermatozoa protein concentrations were assayed using Bradford’s method according to the manufacturer’s instructions (BioRad, Hercules, CA, USA). Protein lysates were then stored at −80 °C until further analysis.

### 2.4. Western-Blotting

The expression of proAKAP4 in bull spermatozoa was first analyzed by immunoblotting using specific AKAP4 antibodies against the prodomain of the AKAP4 precursor (anti-proAKAP4 clone 6F12, 4BioDx, Lille, France) and an anti-AKAP4 antibody (clone 7E10, 4BioDx, Lille, France) against an epitope at the C-terminus of the protein. Bull semen, isolated spermatozoa, and cryopreservative plus the diluted seminal plasma were loaded onto SDS-PAGE.

After determining protein concentrations by the Bradford method, an equivalent of 25 µg of total protein was diluted to 1 volume of 2× concentrated NuPAGE LDS Sample Buffer (ThermoFisher, Waltham, MA, USA) added to 10 µL of NuPAGE Sample Reducing Agent (ThermoFisher, Waltham, MA, USA). Samples were vortexed and heated at 80 °C for 10 min. The whole volume was loaded on polyacrylamide gel (4–12% NuPage Precast Gels) and run for up to 45 min under constant tension of 100 volts per gel. Using the Liquid Transfer System (Life Technologies, Waltham, MA, USA), the gel was transferred onto a 0.45 µm nitrocellulose membrane according to the manufacturer’s instructions (G&E Healthcare, North Richland Hills, TX, USA).

The membrane was then incubated at 4 °C overnight with the first antibody at a dilution of 1:4000 in 25 mM Tris-HCl (pH 8.0), 150 mM NaCl, 0.1% (*v*/*v*) Tween 20 (TBS-T Buffer), either with the clone 7E10, a monoclonal antibody anti-AKAP4 (4BioDx, Lille, France), or with the clone 6F12, a monoclonal antibody anti-proAKAP4 (4BioDx, Lille, France). After appropriate washing steps (3 times 10 min in TBS-T), the membrane was then incubated with a secondary anti-mouse antibody coupled to horseradish peroxidase at 1:50,000 dilution (Vector Laboratories, Burlingame, CA, USA) and further revealed with the ECL™ chemiluminescence kit (G&E Healthcare, North Richland Hills, TX, USA). Images were acquired using the Image Quant™ LAS 4000 system (G&E Healthcare, North Richland Hills, TX, USA).

### 2.5. The Bull 4MID^®^ ProAKAP4 ELISA Assay to Quantify Sperm proAKAP4

A volume of 50 µL of thawed semen sample was mixed with 450 µL of the Bull Lysis Buffer and then processed for ELISA quantification using the Bull 4MID^®^ Kit (4BioDx, Lille, France) according to the manufacturer’s instructions. A quantity of 100 µL of lysates was put into each well of the anti-proAKAP4 antibody-coated plate. A secondary horseradish conjugated proAKAP4 antibody was then added to achieve the sandwich ELISA step. After suitable washing, the substrate solution was added, and coloration was stopped with the stop solution. Since color intensity is proportional to the amount of proAKAP4 present in each semen sample, optical density was measured by spectrophotometry at 450 nm. A standard curve was determined in parallel for precise concentrations of proAKAP4 in the bull semen samples. Results of proAKAP4 concentrations were always expressed in ng/mL or in ng per millions of spermatozoa (ng/M spz) when the spermatozoa concentrations in millions per mL were determined.

### 2.6. Calculation of Non-Return Rate at 90 Days after Artificial Insemination

The non-return rate at 90 days after artificial insemination is the percentage of cows who have not returned to heat after the last artificial insemination. Data were retrospectively collected by Evolution XY (Blain, France) from regular customers who have used these straws to inseminate cows. The threshold was established at 48.8% which is the mean of all groups, and cows with a higher percentage were considered more fertile and cows with a lower percentage were considered less fertile.

### 2.7. Statistical Analysis

Statistical analyses were performed using Prism 8.2 GraphPad software (GraphPad Software, San Diego, CA, USA). Gaussian functions were used to assess the distribution of data. Shapiro Wilk normality tests were applied. The Pearson correlation coefficients were obtained for each condition and thresholds for statistical significance were set at *p* < 0.05. For the non-Gaussian distributed values a Spearman correlation test was applied. In normally distributed groups of data, results were presented as mean ± standard deviation. The significant differences were determined by a non-parametric paired samples Mann–Whitney *U*-test. For non-Gaussian groups of data, median variations were assessed by the non-parametric Friedman test and Wilcoxon signed-rank test.

## 3. Results

### 3.1. Analysis of AKAP4 and proAKAP4 Expression in Post-Thawed Isolated Spermatozoa

The expression of precursor proAKAP4 (or full-length AKAP4) and AKAP4 in bull semen samples was first evaluated by Western blotting. The anti-proAKAP4 clone 6F12 antibodies revealed a single band at 100 kDa corresponding to the proAKAP4 isoform in the isolated spermatozoa. In addition to the 100 kDa band, an additional band at 18 kDa was detected in isolated spermatozoa, corresponding to the cleavage product, also called the prodomain (Figure 1A).

Immunostaining with the AKAP4 antibody clone 7E10 (Figure 1B) revealed two bands, one band at 100 kDa and a more intense band at 82 kDa corresponding, respectively, to the AKAP4 precursor and the mature protein lacking the 182 aa of the prodomain (Figure 1A). Those two bands were observed in isolated spermatozoa but never detected in the cryopreservative alone and in seminal plasma. The precursor proAKAP4 and the mature AKAP4 are therefore specific spermatozoa cell markers and not found in seminal plasma.

### 3.2. ProAKAP4 Concentrations and Total Sperm Motility

Post-thaw sperm motility descriptors are reported in Table 1. Progressive motility ranged between 16% to 53.3% with median at 38.62 and mean ± SD of 38.67 ± 8.55 (*n* = 40). ProAKAP4 concentrations were comprised between 19.96 to 96.95 ng/10 M spermatozoa showing then an almost fivefold amplitude of variations between the post-thawed semen samples (*n* = 40).

The distribution of proAKAP4 concentrations around the mean was rather homogenous as the mean and median were close (Table 1). ProAKAP4 concentrations in post-thawed semen were positively correlated to the progressive motility (*n* = 40, *r* = 0.44, *p* = 0.001) when considering each single ejaculate (Figure 2A). However, to determine whether such a correlation could be indicative of the semen quality of each bull (*n* = 10), the mean of the concentrations of proAKAP4 in ng/10 M of spermatozoa of each bull was compared to the average progressive motility (Figure 2B).

A significant correlation was observed (*n* = 10, *r* = 0.80, and *p* = 0.007) with a calculated linear gain of progressive motility of 5% every 10 ng of proAKAP4 over the concentration of 18.5 ng/10 M of spermatozoa (Figure 2A,B). Assessing the sperm motility descriptors, proAKAP4 concentrations were shown to correlate with the linear and straightness motion of spermatozoa (*n* = 40, rLIN = 0.39, *p* = 0.0006 and rSTR = 0.39, *p* = 0.0001) (Figure 3A,B). ProAKAP4 concentrations in post-thawed semen were also shown to correlate with the beat cross frequency (BCF, *n* = 40, *r* = 0.37, *p* = 0.015). No correlations were found between proAKAP4 concentrations and VCL, VAP, or VSL.

### 3.3. proAKAP4 Dosage and Fertility Rates

Fertility rates were given by the average measure of the non-return rates (NRR in %) at 90 days for each of the 10 bulls (*n* = 199, 100 artificial inseminations with these bulls) and for straws of the same batches than those assessed for proAKAP4 concentrations using the Bull 4MID^®^ ELISA Kit. The mean NRR was calculated and was of 48.8% ± 0.045. This mean was used to determine how proAKAP4 concentrations were distributed between ejaculates with NRR values below or over 48.8%. As shown on Figure 4, proAKAP4 concentrations were significantly higher in ejaculates from bulls giving NRR over the threshold of 48.8% and the average concentration of 52.14 ± 3.38 ng/10 M of spermatozoa whereas the proAKAP4 average concentration was of 39.55 ng/10 M of spz for all ejaculates that gave NRR below 48.8%.

## 4. Discussion

In our study we demonstrated for the first time a correlation between proAKAP4 concentrations and bull fertility. Several studies [29,30,31,32,33] previously identified AKAP4 as one of the genes, transcripts, or proteins most expressed in fertile cattle, mainly by proteomics methods. In our study, the fertility rate was expressed by non-return rate (NRR) at 90 days after artificial insemination, as this rate discriminates non-fertile insemination, early and late embryonic loss, and early abortion. A high correlation was obtained between proAKAP4 concentrations and NRR-90, indicative of a positive association with an increased rate of fertility. With the threshold of 48.8% as the NRR mean we showed that bulls above this mean had significantly higher proAKAP4 concentrations than those placed below. Even if other techniques such as the conception rate can predict fertility more specifically, the non-return rate is the easiest data to collect when working with large numbers of breeding cows and where access to certain data (rank and stage of lactation, pregnancy checks, etc.) is not possible [34]. The non-return rate is the predominant phenotype for assessing male reproductive status in the field [35,36] confirming a pregnancy and working towards the dairy industry objective, which is to obtain a live calf.

Spermatozoa are highly differentiated cells whose principal function is to fertilize an oocyte. To achieve this function, spermatozoa have to reach and penetrate the oocyte, with the aid of tail motility. Thus motility has been widely used as an indicator of bull fertility since immotile spermatozoa cannot fertilize the oocyte [14]. Miki et al. [18], and later Fang et al. [20] and Xu et al. [21], have shown that mice models engineered to be deprived of the AKAP4 gene were infertile. Carr and Hanlon Newell [37] determined AKAP4 to be a key molecule regulating the motion functions of flagella and proAKAP4 was a specific protein of the fibrous sheath of spermatozoa [17]. Spermatozoa motion was then shown to be impaired in mice lacking the AKAP4 gene [18,20,21]. As in other mammals such as horses, mice, and even camels, proAKAP4 quantity in spermatozoa is indicative of the sustained quality and functionality of spermatozoa [25,27]. In our study, the strong correlation between progressive motility (r = 0.80) of each individual bull was found to be in agreement with the results reported in stallion and human sperm, and recently in Simmental bull semen [23,24,26]. Several parameters of motility correlated with the concentration of proAKAP4, for instance, the linear (VSL), curvilinear (VCL), and average (VAP) velocity. In addition, the progressive motility, linearity, and the frequency of increase in trajectory were related to the protein concentrations, suggesting that the amount of proAKAP4 was likely related to progressive motility and the flagellum beat frequency associated with a linear and straight motility. However, according to studies by Farrell et al. [38] and Nagy et al. [39] VAP, VCL, VSL, BCF, and LIN are correlated with fertility. The same is true for progressive motility in Puglisi et al.’s study [40]. The proAKAP4 concentration taken with motility descriptors provides more information about sperm quality, and predicts male fertility and the performance of ejaculate conditioned in straws [27]. Singh et al. [30] have reported previously that AKAP4 transcripts were differentially expressed (*p* < 0.01) among bull semen samples of high and low quality. ProAKAP4 was therefore indirectly correlated with fertility for all ejaculates through motility parameters. Similar conclusions were deduced in stallions where the proAKAP4 concentrations were correlated with progressive motility, as a predictive marker of fertility in the stallion [24]. Further analyses are needed to relate proAKAP4 concentrations with sperm motile subpopulations to describe better the relationship between proAKAP4 and motion parameters. A faster and a more motile subpopulation of spermatozoa having the greatest effect in vitro yield embryo followed by a subpopulation consisting of hypermotile spermatozoa [41], then proAKAP4 is expected to be in a higher concentration than in other subpopulations.

Furthermore, proAKAP4 is not only a motility sperm quality marker, but also a spermatic maturation indicator. The AKAP4 gene belongs to the X-linked member of the AKAP gene family [42]. Both proAKAP4 and AKAP4 protein expression were shown to be strictly restricted to spermiogenesis stages from round spermatids to mature spermatozoa [42,43]. They are consequently described as indicators of good spermatogenesis by being sensors of oxidative damage [43]. Variations of proAKAP4 concentrations may appear as a physiological explanation for the loss of sperm motility and capacitation competence, as commonly met with in oxidative stress conditions. The degradation of proAKAP4 may then be regarded as a natural protective mechanism that limits the transmission of an altered male genome to the next generation. In this sense, proAKAP4 concentrations represent an in vitro functional sperm quality assay. Interestingly, AKAP4 was highly represented in the spermatozoa of fertile compared with infertile men that had lost the capacity to bind in vitro to the zona pellucida [44]. AKAP4 was also identified among the capacitation-associated tyrosine phosphoproteins in buffalo and cattle spermatozoa [45]. Since AKAP4 gene and products are highly conserved among mammals and described in cattle (Holstein and Simmental), buffalo, and yaks [23,30,31,46], this functional approach may be then applied to any bull breeding centers.

Our study was performed on post-thawed semen but could also be clearly undertaken on fresh semen or to compare successively fresh and frozen semen in order to assess whether the freezing step preserves the reserve of motility as represented by proAKAP4. This would be of particular interest in semen collection centers to measure proAKAP4 concentration in ejaculates before cryopreservation. ProAKAP4 was then proposed as a predictive tool for sperm quality evaluation in order to optimize freezing and cooling protocols with artificial insemination settings in rams and in bull breeding centers [47,48,49,50].

These preliminary results suggest that measuring the concentration of proAKAP4 is a promising tool to evaluate the viability and long-lasting functionality of semen and also for the selection of the optimal freezing medium for each bull as reported for stallion semen [47].

Measuring the quantity of proAKAP4 as a sperm quality tool brings a new technology to the artificial insemination industry to evaluate the quality of fresh ejaculates and after thawing as quality control in straw production.

## 5. Conclusions

In conclusion, the data from our study suggest that the sperm protein proAKAP4 can be used as a marker of post-thawed semen quality and of fertility in bulls. In fact, the proAKAP4 concentrations were shown to correlate with progressive motility and to be associated with sperm motion variables and the 90-day non-return rates following cow artificial insemination. As this study was performed with Holstein breed cattle and a limited number of bull ejaculates, further studies will be necessary to extend these results to other bovine breeds, since proAKAP4 sequence and metabolism were shown to be highly conserved.

## Figures and Tables

**Figure 1 vetsci-09-00224-f001:**
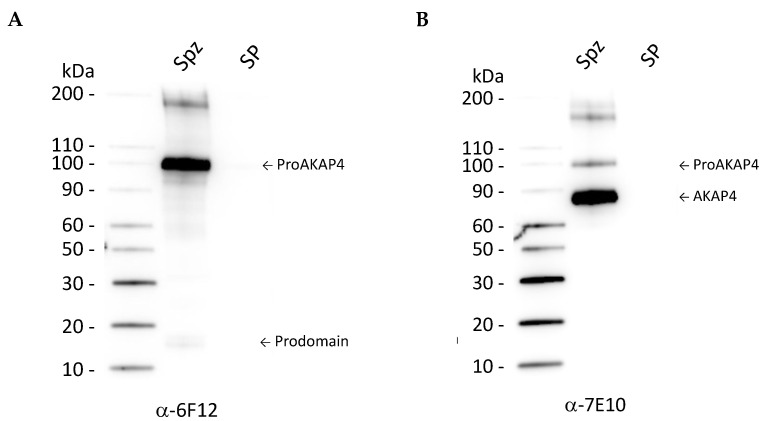
(**A**) Western blot analysis of ProAKAP4 and AKAP4 expression in post-thawed isolated spermatozoa (SPZ) in post-thawed cryopreservative without spermatozoa (SP). ProAKAP4 and the prodomain (indicated by arrows) were detected in the spermatozoa (SPZ) lane at 100 and 18 kDa but not in the extender (SP) diluted with the cryopreservative (SP) with the 6F12 mouse monoclonal purified antibody. (**B**) Both proAKAP4 and AKAP4 were detected at 100 and 82 kDa with the 7E10 mouse monoclonal purified antibody. No labelling was observed in the lane with extender with cryopreservative without spermatozoa (SP lanes). Apparent molecular weights were indicated on the left of the SDS-PAGE lanes and determined using calibrated molecular weight markers. The whole Western blot figure in the Appendix A.

**Figure 2 vetsci-09-00224-f002:**
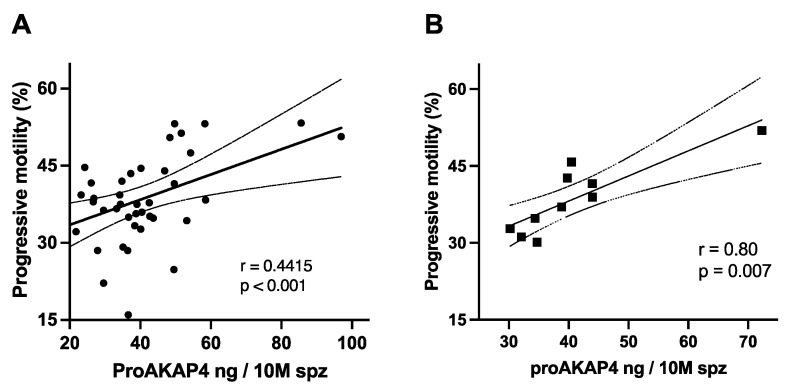
Correlation between proAKAP4 concentration and progressive motility in post-thawed bull semen. (**A**) Correlation of each individual straw (**B**) Correlation of the average of all straws of the same bull. ProAKAP4 concentration in ng per 10 million of spermatozoa (ng/10 M spz) of spermatozoa and percentage of progressive motility.

**Figure 3 vetsci-09-00224-f003:**
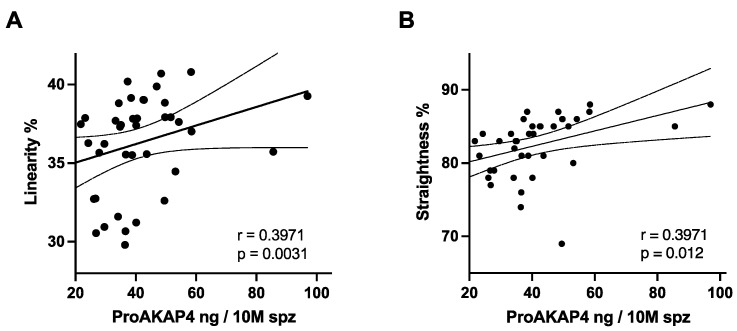
Correlation of linearity (**A**) and straightness (**B**) with proAKAP4 concentrations in post thawed bull semen. (**A**) Linearity (**B**) straightness of spermatozoa and concentrations of proAKAP4 in ng par 10 million of spermatozoa (ng/10 M spz).

**Figure 4 vetsci-09-00224-f004:**
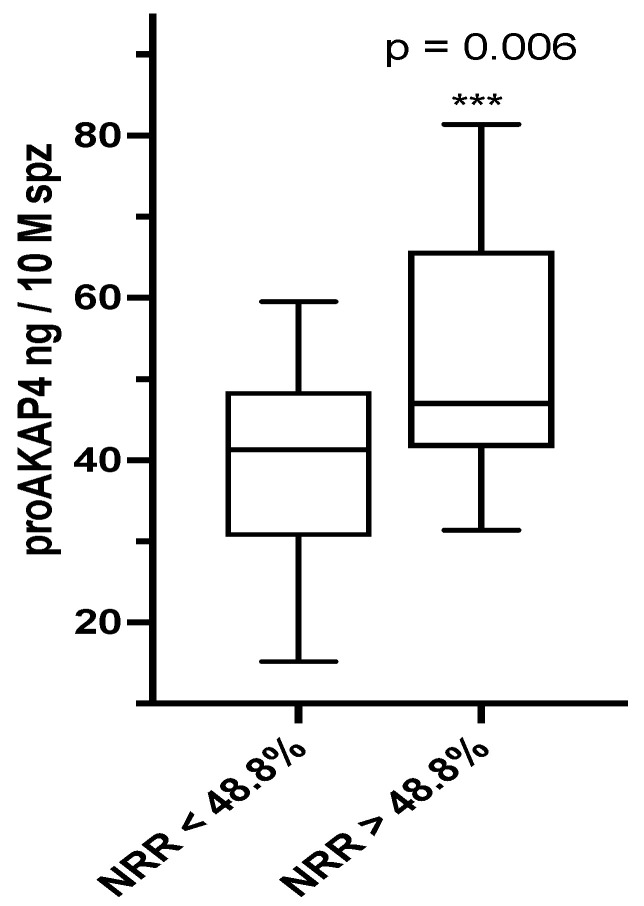
Box representation of the proAKAP4 average concentrations in post-thawed straws that gave a NRR% at 90 days over or below the threshold of 48.8%. NRR: non-return rate at 90 days post insemination and proAKAP4 concentration as a ng per 10 million of spermatozoa (ng/10 M spz). *n* = 199, 100 artificial inseminations. *** means significant differences (*p* = 0.006).

**Table 1 vetsci-09-00224-t001:** Descriptive statistics of proAKAP4 values and all motility descriptors. Minimum and maximum values (min. and max.), the 25, 75 percentile and median, the mean and standard deviation (SD), as well as the lower and upper 95% confidence intervals (CI). ProAKAP4 ng/10 million of spermatozoa (ng/10 M spz), Progressive motility (PR), average path velocity (VAP), straight line velocity (VSL), curvilinear velocity (VCL), beat cross frequency (BCF), linearity (LIN), and straightness (STR).

	proAKAP4 ng/10 M spz	PR (%)	VAP (μm/s)	VSL (μm/s)	VCL (μm/s)	BCF (Hz)	LIN (%)	STR (%)
** *n* **	**40**	**40**	**40**	**40**	**40**	**40**	**40**	**40**
**Min.**	19.96	16.00	77.65	53.78	142.5	29.80	32.83	69.00
**25%**	30.51	34.46	92.24	77.57	162.6	34.73	42.30	80.00
**Median**	38.62	37.92	98.94	84.06	175.1	37.36	44.83	83.00
**75%**	49.23	44.38	105.0	88.37	187.8	38.60	48.46	85.00
**Max**	96.95	53.33	131.5	103.5	251.7	40.80	53.17	88.00
**Mean**	41.07	38.67	100.9	83.20	181.6	36.28	44.87	82.40
**SD**	15.42	8.554	12.45	10.21	26.77	3.048	4.516	4.088
**Lower 95% CI**	36.14	35.93	96.96	79.93	173.0	35.30	43.42	81.09
**Upper 95% CI**	46.00	41.40	104.9	86.46	190.2	37.25	46.31	83.71

## Data Availability

MDPI Research Data Policies at https://www.mdpi.com/ethics. Accessed on 3 May 2022.

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
