# Peer review of "ProAKAP4 Semen Concentrations as a Valuable Marker Protein of Post-Thawed Semen Quality and Bull Fertility: A Retrospective Study"

_vetsci, 2022, doi:10.3390/vetsci9050224_

Round 1
Reviewer 1 Report
General comments
The authors have looked at an innovative parameter of semen and its correlation with bull fertility, as well as with other parameters traditionally used in semen assessments. However, the number of bulls is very low, the stats and the language need some work. As written, the authors’ point is sometimes lost.
Given the availability of samples for domestic species, why were only 10 bulls used?
Why was this study not performed on fresh semen first rather than suggesting this be done at the end? It seems that it would have been good to do that since there are so many other factors can affect motility after freeze-thaw.
In order to demonstrate the value of the paper, the following point needs to be clarified in the introduction: If proAKAP4 is directly correlated with motility but must be analyzed in a lab, what is the advantage of performing this (costly and time consuming) process over assessing motility bull-side?
Abstract
The abstract is written as though everyone knows what AKAP4 is. This is not the case, so a sentence explain what the function of this protein is, would help readers not as familiar with the molecular aspects of semen see what the importance of the work is.
Background should also include some text about how freezing/ thawing affects AKAP4, these freeze thaw cycles do affect sperm viability.
Figure 1
The labels are obscuring the figure- need to reformat.
It is unclear to me why the authors test for normality, but then report both non-parametric AND parametric descriptive statistics.
Table 1
It looks like statistically the 4 straws for the 10 bulls were analyzed as independent variables, when in fact what the authors have is an n=10 with 4 replicates
How were the nearly 200K inseminations distributed over the 10 bulls? How were they distributed over the 4 batches that were analyzed for each bull?
It also seems that these should be analyzed as replicates rather than independent samples- they are replicates of each batch which are replicates of each bull.
It would be good to explicitly expand on how NRR at 90d discriminates infertile AI from EED? How would this differentiate between infertile AI and death of embryo within the first 30d? What are other options (such as the NNR at 56d) and what are advantages/ disadvantages of these?
Please provide some context for NRR, just under a 50% chance of pregnancy seems pretty low in an industry that depends on pregnancies for production. Particularly in light of the fact that NRR can vary a lot between bulls- and the fact that other publications include NRR much higher than this.
Since inseminations were not part of a controlled trial there would be a large amount of variation introduced with each shipment, each person handling and thawing out semen, and each insemination. This would be valuable to add to the discussion
Language:
Language needs revision, some of the sentences are incorrectly constructed. Preposition use is also often incorrect as well.
Author Response
To reviewer 1:
General comments
The authors have looked at an innovative parameter of semen and its correlation with bull fertility, as well as with other parameters traditionally used in semen assessments. However, the number of bulls is very low, the stats and the language need some work. As written, the authors’ point is sometimes lost.
First of all, we are grateful to the referees for their comments and concerns We understand that we need to improve some points and we will do our best to bring more clarity to the stats and improve the language levels. Then two independent native English speakers will correct the manuscript as suggested by referees. Within the scope of this study, our main objective was to evaluate the relationship of proAKAP4 concentrations with sperm parameters of post-thawed straws (n=2 per batch of straw) from 4 individual batch of straw produced between year 2011 and 2017 of Holstein bulls (n=10) from EVOLUTION which is producing semen doses and have all the fertility data (France). This is then clearly a retrospective study from existing stocks of Holstein bull straws that were used for artificial insemination (AI), and for which the total non-returned rate (NRR) were collected from different farms during the same period, and made available for our present study.
Q1: Given the availability of samples for domestic species, why were only 10 bulls used?
Answer: The number of Bulls included in the present study is related to the design of the study. We use commercialized straws produced between year 2011 and 2017 of Holstein bulls that were still available at the time of the study from the breeding center and for which the NRR were known.
Q2: Why was this study not performed on fresh semen first rather than suggesting this be done at the end? It seems that it would have been good to do that since there are so many other factors can affect motility after freeze-thaw.
Answer: In a first attempt on this newly commercialized functional marker, we made the choice to evaluate the proAKAP4 concentration, therefore the quality of the straws that are currently commercialized based on classic semen analysis parameter
Analysis of proAKAP4 concentrations in collected fresh ejaculates is highly interesting but for another objective that will be to optimize or select semen before straw processing based on a molecular marker. . The relationship between fresh ejaculate quality before and after straw production is out of the scope of the present study and should be addressed in another research investigation with a different experimental design.
Q3: In order to demonstrate the value of the paper, the following point needs to be clarified in the introduction: If proAKAP4 is directly correlated with motility but must be analyzed in a lab, what is the advantage of performing this (costly and time consuming) process over assessing motility bull-side?
Answer:
ProAKAP4 spermatozoa concentration ensure, after proteolysis, the amount of AKAP4 that is structurally associated to the fibrous sheath of the principal piece de spermatozoa’s flagellum to enable the transduction of signals regulating long-term motility, hypermotility, capacitation and fertility ProAKAP4 is e associated to the long-term motility proper maintenance overtime. As shown in other species (Horse, Camel …), we show in this manuscript that proAKAP4 concentrations correlate to post-thawed semen progressive motility and motion parameters related to linear motility, known to be essential for fertility. In contrast, CASA parameters taken alone were not correlated with the fertility results in Holstein Bulls and therefore not sufficient to predict fertility outcomes. In sharp contrast, proAKAP4 concentrations correlate with sperm parameters and the Bull fertility results. There is then an added value to consider proAKAP4 as a functional marker of Holstein bull semen quality, long-lasting functionality and predictive of Bull fertility herein based of post-thawed proAKAP4 concentrations.ProAKAP4 provide an independent and valuable parameter of the Bull semen quality produced in AI centers. The need of a laboratory is overstated since all the material needed stands on a single bench or table, and does not need high technicity. The ELISA method for proAKAP4 quantification allow to process 88 samples at the time (and up to 5 plates at the time then 440 samples at the time) in less that 3 hours and appears then an easy and time saving approach for semen control quality assessments compared to CASA and microscopic evaluations.
Abstract
Q4: The abstract is written as though everyone knows what AKAP4 is. This is not the case, so a sentence explain what the function of this protein is, would help readers not as familiar with the molecular aspects of semen see what the importance of the work is.
Answer: The abstract has been modified to add a sentence about proAKAP4 and AKAP4 main known functions.
Q5: Background should also include some text about how freezing/ thawing affects AKAP4, these freeze thaw cycles do affect sperm viability.
Answer: The background has been modified, the effect of freezing/thawing cycles were published in Dewulf Q, Brian-Amirat L, Eddarkaoui S, Chambonnet F, et al. The Effects of Freeze-Thaw Cycles and of Storage Time on the Stability of Proakap4 Polypeptide in Raw Sperm Samples: Implications for Semen Analysis Assessment in Breeding Activities. Dairy and Vet Sci J. 2019; 13(3): 555861. DOI: 10.19080/JDVS.2019.13.5558610) Furthermorewe assessed the proAKAP4 concentration from one straw of the same ejaculate and after freeze/thaw at -20°C to verify the biochemical stability after 5 freeze/thawing cycles. As previously reported, ProAKAP4 is stable after several freeze/thawing cycles. Change in proAKAP4 concentrations in post-thawed semen added with a cryopreservative and conditioned in straws is not related to proAKAP4 biochemical instability. Modification of proAKAP4 concentrations is therefore related to the regulation of the proAKAP4 expression level during spermatogenesis, to the conversion of proAKAP4 precursor into AKAP4, or the degradation of the protein resulting from environmental stress (Nixon et al., 2019).
Figure 1
Q6: The labels are obscuring the figure- need to reformat.
Answer: on the revised manuscript the figure has been reformatted.
Q7: It is unclear to me why the authors test for normality, but then report both non-parametric AND parametric descriptive statistics.
Answer: Non-parametric statistic tests are recommended when values are not distributed following a normal Gaussian bell-shape distribution. In contrast, the normal Gaussian value distribution allows the use of parametric statistical tests. This is why normality tests were used by the authors before to choose the adequate statistical tests. As mentioned in the manuscript, the normal distribution test used was the Shapiro-Wilk test.
Table 1
Q8: It looks like statistically the 4 straws for the 10 bulls were analyzed as independent variables, when in fact what the authors have is an n=10 with 4 replicates
Answer: For each of the 10 bulls, 2 straws of each batch for a total of 4 different batches were used both for sperm parameter analyses and proAKAP4 concentrations. The mean concentration of the 2 straws of each individual batch was considered as an independent value since straws produced in the 4 individual batches were obtained from 4 independent ejaculates. Therefore, in total 4 values per bull (40 values, as indicated on Table 1) for proAKAP4 concentration were considered. These values are not replicates since there are not from the same bull ejaculate with sometimes more than one year difference between the times of batch production
Q9: How were the nearly 200K inseminations distributed over the 10 bulls? How were they distributed over the 4 batches that were analyzed for each bull?
Answer: Were included in the present study, 10 Bulls for which a minimum of 2 straws of 4 individual batch of straws – meaning that they were produced from 4 different collected bull’s ejaculates – and for which the TNR at 90 days were available between the period of 2011 to 2017. This is a retrospective study. Although not included in the manuscript please find the summary of the TNR90 recovered from the 10 bulls within the period between 2011 to 2017.
Total number IA TNR90 mean SD
9525608 49% 0,04481186
Number IA TNR90 SD
GARCILL 216 60,18% 11,49%
GARONET 103 55,34% 6,65%
HELMONDO 22392 50,02% 1,33%
HORTUN ISY 11861 49,14% 0,45%
FRENES 18682 49,09% 0,41%
GALOSH ISY 13352 48,40% 0,29%
HOREDP ISY 4510 48,35% 0,34%
GEDELOIR 12127 47,99% 0,70%
FINGER 37443 47,47% 1,22%
FYJNOE ISY 16255 46,45% 2,24%
HOTELOT 24799 44,44% 4,25%
FUMPOO ISY 21444 44,40% 4,29%
HADANGE 15916 44,28% 4,40%
Total 199100 48,89% 2,93%
Between the period comprised between 2011 and 2017, a total of 199100 IA and TNR90 were recovered from the 10 Bulls included in our study.
This table was provided as the raw data used for our study and can be added as supplementary results. In that case we want to preserve the anonymity of the bulls used in the study and replace their name with bull 1,2,3 ….10.
Q10: It also seems that these should be analyzed as replicates rather than independent samples- they are replicates of each batch which are replicates of each bull.
Answer: Each value (sperm parameters, motion parameters and proAKAP4 concentrations) from 4 individual ejaculates of the same bull were not considered replicates since they are 4 different biological samples collected separately. 40 values were therefore considered for proAKAP4 concentrations.
Q11: It would be good to explicitly expand on how NRR at 90d discriminates infertile AI from EED? How would this differentiate between infertile AI and death of embryo within the first 30d? What are other options (such as the NNR at 56d) and what are advantages/ disadvantages of these?
Answer: The non-return rate represents the number of females inseminated in primary insemination for which a recall of insemination is not requested. The cow has not returned to heat since insemination, so it is believed that the animal was fertile and does not require further AI. NRR are calculated on three dates after insemination: 30 days, 56 days and 90 days. The 30-days no-return rate will rule out unfertilizing AI and early embryonic mortality.
This is not easy to discriminate against the non-fertilization of an early embryonic mortality. NRR56-d will make it possible to exclude in addition cases of late embryonic mortality. Finally, NRR90 will exclude abortions from early gestation. NRR depend on many male and female factors, with NRR-90 being the most discriminating, making it the most reliable parameter for studying male fertility. However, there are some biases in this evaluation system which imply that some animals are wrongly taken into account in the calculation of the NRR, as Culled cows will not have an insemination recall, even if it has not been fertilizing and Cows that are bred by the bull after insemination failure. Finally, it should be noted that the NRR obtained can be corrected by taking into account the season, the age of the inseminated animal, the herd and the technician.
Q12: Please provide some context for NRR, just under a 50% chance of pregnancy seems pretty low in an industry that depends on pregnancies for production. Particularly in light of the fact that NRR can vary a lot between bulls- and the fact that other publications include NRR much higher than this.
Answer: It’s true but that are the results transmitted from the AI centers concerning the bulls used in our study, NRR was of 48,89% on average with a minimum of 44,28%, a maximum 60,18% and a mediane at 48.4%, meaning that 5 NRR were over the value of the mediane and 5 NRR were below this value. The NRR are not representative of the overall performance of breeding centers or the average NRR obtained for all Bulls at EVOLUTION. For the present study, a sufficient amplitude of NRR values were needed for statistical analyses, which also explain the low number of Bull included in the study. We agree that this pioneering study should be complete in the future with a larger number of Bull semen samples that will enable to assess more restricting intervals of NRR values.
A variation in NRR was observed according to season and photoperiod only for beef breeds. For the Holstein breed, in our very temperate region, we observe a decrease in spring and autumn, probably related to the conditions of breeding of the females (pasture/ stable). To the best of our knowledge, there is no seasonal variation according to the month of production for males in Hostein breed.
In addition of these information, EVOLUTION AI Center has its own indicators that have been corrected but are not available in this trial for confidentiality reasons. To overcome this constraint, the only fertility correction was the parity of the females. NRR-90d heifers were ruled out and only NRR-90d cows, more discriminating were used in the present study. Furthermore, the data relate only to Holstein’s breed. Finally, we are talking about at least more than 11800 AI on cows for each bull, which brings a very high reliability to the NRR values used. The correlation of T NRR-90d cows in Holstein breed with fertility can therefore be considered strong in view of Evolution experience.
Q13: Since inseminations were not part of a controlled trial there would be a large amount of variation introduced with each shipment, each person handling and thawing out semen, and each insemination. This would be valuable to add to the discussion
Answer: We partially agree, as are other parameters and factors related to the cow itself, the breeder, heat detection and feeding factors of the female, but as indicated above this is a retrospective study, and the inseminator part was not taken into account in this study. Nevertheless, the EVOLUTION AI center works with experienced inseminators who are subjected to procedures as part of the quality control set up by the company. Therefore, we are likely confident that NRR differences, while considering a large number of AI, is related to the Bull semen quality rather than other parameters such as the age of the Bull, which we showed as not being a confounding statistical parameter.
Language:
Q14: Language needs revision, some of the sentences are incorrectly constructed. Preposition use is also often incorrect as well.
Answer: The document has been submitted for proofreading and correction by two native English speakers.
Reviewer 2 Report
The introduction is described sufficiently and introduces the reader to the topic undertaken by the researchers.
Materials and methods.
2.1.
The number of bulls used in this study is too low.
Please add information (table) about the age of each bull or group of bulls if they are of the same age.
Please add information about how animals were kept. Did they have the same environmental conditions?
Please precise „different days”. The research confirms the influence of the breeding method and the season of the year on the quality of semen and also factors such as nutrition, housing conditions, and intensity of the breeder's exploitation are important for the quality of the sperm.
The results are presented logically and allow the reader to follow the course of the analysis. The tables and figures are informative. All calculations were carried out with the use of appropriate statistical methods and were thoroughly described and presented in the results.
In the discussion, the authors discussed the results of the research in relation to the literature and touched upon the most important problems arising from the research. The discussion is written in a way that is understandable to the reader.
In my opinion, the work is written in the correct language and understandable to the reader.
Author Response
Reviewer 2:
The introduction is described sufficiently and introduces the reader to the topic undertaken by the researchers.
Materials and methods.
2.1.
Q1 The number of bulls used in this study is too low.
Answer It should be noted that this is a retrospective study based on existing stocks of semen for bulls that have been used and have an NRR. The low number of bulls is due to these 2 constraints plus another related to the optimization of stocks which means that there are few bulls left with available stocks.
Q2 Please add information (table) about the age of each bull or group of bulls if they are of the same age.
Answer: The table with the age of each bull at semen collection was provided with the raw data. This table can be added with the supplementary material. Ages of the bull were compared with semen parameters, motion parameters, and proAKAP4 concentrations and were not significantly correlated with any of the parameters (Use of an ANOVA statistic test), therefore suggesting that aging is not a confounding parameter in the present study.
Straw batch date Age (months)
FUMPOO ISY 28/08/2015 67
28/09/2015 68
28/10/2015 69
23/11/2015 70
HORTUN ISY 02/05/2014 25
15/10/2014 30
03/11/2014 31
01/12/2014 32
HOREDP ISY 08/12/2014 33
29/12/2014 33
22/04/2015 37
07/05/2015 38
GALOSH ISY 08/02/2016 57
25/02/2016 58
27/06/2016 62
18/07/2016 62
HELMONDO 26/06/2014 30
04/07/2014 30
15/10/2014 33
27/10/2014 34
FYJNOE ISY 21/10/2011 21
21/11/2012 34
12/12/2012 35
30/01/2013 37
HOTELOT 13/05/2015 29
11/09/2015 33
16/10/2015 34
13/11/2015 35
GEDELOIR 16/01/2017 67
08/02/2017 69
16/03/2017 65
26/04/2017 66
FRENES 09/06/2016 70
29/07/2016 72
26/08/2016 73
30/09/2016 74
GARONET 26/04/2013 17
27/05/2013 19
27/06/2013 20
25/07/2013 20
FINGER 28/10/2015 67
27/11/2015 68
28/12/2016 81
30/01/2017 82
HADANGE 06/02/2015 31
02/03/2015 32
23/03/2015 32
08/04/2015 33
GARCILL 10/07/2017 70
21/07/2017 70
31/07/2017 70
16/08/2017 71
Q3 Please add information about how animals were kept. Did they have the same environmental conditions?
Answer: The animal had the same food and environment conditions at the AI center. We then added it in the method section. (à mettre dans le manuscript)
Q4 Please precise different days”. The research confirms the influence of the breeding method and the season of the year on the quality of semen and also factors such as nutrition, housing conditions, and intensity of the breeder's exploitation are important for the quality of the sperm.
Answer: We agree but this is a retrospective study and we have adapted to the straw stock available at the AI production center for which NRR were available and validated through other internal studies, which we hope you will understand, these studies are confidential. Concerning different days, refer to the table (Q2). The age values are in months and related to the season of semen collection and straw production. No significant relationship was observed between the age of the animal at the date of semen collection and any of the parameters considered in the present study.
The results are presented logically and allow the reader to follow the course of the analysis. The tables and figures are informative. All calculations were carried out with the use of appropriate statistical methods and were thoroughly described and presented in the results.
In the discussion, the authors discussed the results of the research in relation to the literature and touched upon the most important problems arising from the research. The discussion is written in a way that is understandable to the reader.
In my opinion, the work is written in the correct language and understandable to the reader.
Reviewer 3 Report
The article shows that there is a relationship between the concentrations of ProAKAP4 in semen with the quality of semen and its rate of non-return of inseminated cows. The methodology is clear and the results of the study are useful to use this marker as an indicator of potential fertility of bulls used in artificial insemination. 10 bulls were used and 4 extractions were made to each one, it is not clear if there was a selection of the ejaculates that were later frozen. Another question is whether it is possible to identify the relationship of ProAKAP4 with a particular ejaculate or if it is associated with a particular bull?
Author Response
Reviewer 3
Comments and Suggestions for Authors
The article shows that there is a relationship between the concentrations of ProAKAP4 in semen with the quality of semen and its rate of non-return of inseminated cows. The methodology is clear and the results of the study are useful to use this marker as an indicator of potential fertility of bulls used in artificial insemination.
Q1:10 bulls were used and 4 extractions were made to each one, it is not clear if there was a selection of the ejaculates that were later frozen.
Answer Regarding ejaculates, we have used the selected ejaculates that were frozen. There is a variability between batches that can be related to the condition of the bull at the time of collection. But as ejaculates are sorted, only those with sufficient quality are kept.(In total, 10-12% of all breeds are eliminated, less than 10 for dairy breeds and higher for beef breeds).
Q2:Another question is whether it is possible to identify the relationship of ProAKAP4 with a particular ejaculate or if it is associated with a particular bull?
Answer: As you can see in figure 2, both are possible. A correlation is observed between increased proAKAP4 concentrations and progressive motility of each separate ejaculated. A similar correlation between averaged concentrations of proAKAP4 of the 4 ejaculates of the same Bull and both progressive motility and total motility suggesting that proAKAP4 concentrations is indicative of Bull semen quality for each separate Bull ejaculate and can also provide the semen of quality overall produced by a single Bull when proAKAP4 concentrations are measured for each collected and pro cessed ejaculated and averaged. Therefore, proAKAP4 on a single ejaculate (meaning here post-thawed straw) is not sufficient to summarize the semen quality that will be produced by a single Bull. A follow-up of each batch of straw produced from collected ejaculate should be performed.
Round 2
Reviewer 2 Report
Dear Authors
All my previous suggestions have been introduced into the manuscript. Now, I am content with the new version of this manuscript.
Kind regards
Reviewer
Author Response
Dear reviewer,
Thanks for you effort and your kind review.
